# Travel-Related Monkeypox Outbreaks in the Era of COVID-19 Pandemic: Are We Prepared?

**DOI:** 10.3390/v14061283

**Published:** 2022-06-13

**Authors:** Oyelola A. Adegboye, Maria Eugenia Castellanos, Faith O. Alele, Anton Pak, Henry C. Ezechukwu, Kay Hou, Theophilus I. Emeto

**Affiliations:** 1Public Health & Tropical Medicine, College of Public Health, Medical and Veterinary Sciences, James Cook University, Townsville, QLD 4811, Australia; maria.castellanosreynosa@jcu.edu.au (M.E.C.); faith.alele@jcu.edu.au (F.O.A.); theophilus.emeto@jcu.edu.au (T.I.E.); 2World Health Organization Collaborating Center for Vector-Borne and Neglected Tropical Diseases, College of Public Health, Medical and Veterinary Sciences, James Cook University, Townsville, QLD 4811, Australia; 3Australian Institute of Tropical Health and Medicine, James Cook University, Townsville, QLD 4811, Australia; 4Centre for the Business and Economics of Health, The University of Queensland, Brisbane, QLD 4067, Australia; a.pak@uq.edu.au; 5Department of Medical Biochemistry, Eko University of Medicine and Health Sciences, Ijanikin 102004, Lagos State, Nigeria; hezechukwu@ekounimed.edu.ng; 6College of Medicine and Dentistry, James Cook University, Townsville, QLD 4811, Australia; kay.hou@my.jcu.edu.au

**Keywords:** travel medicine, communicable disease control, monkeypox, COVID-19, orthopoxvirus

## Abstract

Several neglected infectious pathogens, such as the monkeypox virus (MPXV), have re-emerged in the last few decades, becoming a global health burden. Despite the incipient vaccine against MPXV infection, the global incidence of travel-related outbreaks continues to rise. About 472 confirmed cases have been reported in 27 countries as of 31 May 2022, the largest recorded number of cases outside Africa since the disease was discovered in the early 1970s.

## 1. Introduction

Monkeypox virus (MPXV) is a re-emerging zoonotic virus caused by orthopoxvirus and results in a smallpox-like condition in humans [1,2]. Since the first human cases of monkeypox were discovered in the Democratic Republic of the Congo (DRC) in 1970, the disease has spread to other Central and West African countries, with cases outside of Africa emerging in recent years [3]. The exportation of cases of monkeypox has been linked to recent outbreaks in Nigeria, which began in 2017 with a total of 88 cases [4]. Subsequently, sporadic cases of monkeypox in Nigeria have been reported in 2018, 2019, 2020, 2021 and 2022 (49, 47, 8, 34 and 21 cases, respectively) [5]. At the time of writing this report, the first death associated with the virus was recorded in Nigeria [5]. The most recent travel-related incidence of MPXV was in the UK in April/May 2022 [6].

Currently, there are multiple outbreaks in Europe, the Americas and Australia [7]. It is important to note that the initial imported cases originated from Nigeria. While travel plays a vital role in infectious disease outbreaks, as seen in the case of COVID-19 [8], interestingly, MPXV has spread across different countries, with more recent cases not related to travel to endemic regions (Figure 1) [9]. The ongoing MPXV outbreak has been reported in 27 countries, and about 472 cases have been confirmed (31 May 2022), thus the largest number recorded outside of Africa since the disease was discovered in the 1970s (Table 1).

Historically, the first outbreak outside Africa was reported in 2003 in the US [18,19,20]; no other travel-related cases were reported until 2018. From 2018 to April 2022, nine travel-related MPXV events have been recorded in four countries (UK, Singapore, Israel, and the US), resulting in at least eleven laboratory-confirmed cases (Table 2). In 2018, three events occurred outside Africa, two in the UK and one in Israel, resulting in four cases of human MPXV; in 2019, two cases from two outbreaks were recorded in Singapore and the UK [4]. There were no travel-related outbreaks reported in 2020. However, three outbreaks were recorded in 2021, one in the UK and two in the US (n = 5 cases) [4].

To better understand the disease, a genomic analysis of ten MPXV isolates was conducted during the outbreak in 2017 and 2018, when the first cases of exported MPX were reported [4]. The samples comprised five local (Nigerian) MPXV cases, four exported cases (UK1, UK2, Israel, Singapore) and one nosocomial transmission case within the UK (UK3) [4]. The study showed that the exported cases during the outbreak belonged to the West African clade and shared a common ancestor with a local case in Bayelsa State with a travel history to Bayelsa, Delta or Rivers State [4].

During this current outbreak, a genomic sequence conducted in Portugal suggests that the 2022 virus belongs to the West African clade and may be related to the 2018 and 2019 MPX viruses exported from the region [17]. However, it is interesting to note that the more recent cases in other parts of the world have not been associated with travel to endemic areas or mapped epidemiologically to West or Central Africa [9]. Given this new development, the disease is now topical, indicating a possible community spread of the disease in an environment where it is not endemic. Based on current available data, 98% (214/217) of confirmed cases who indicated their gender were men [10]. Furthermore, some of the ongoing outbreak clusters have been reported among men who have sex with men [7]. This pattern is considered unusual as there have been no previous reports of possible sexual transmission of the disease [7].

## 2. Impact of COVID-19 on Monkeypox Surveillance in Nigeria

Amid the monkeypox outbreak, the first COVID-19 case in Nigeria was reported in March 2020 [29,30]. The onset of COVID-19 limited international travel and movement, which may account for the few cases of MPVX reported in 2020 and no record of travel-related or exported cases [30]. The focus on COVID-19 may have been a disadvantage in the fight against monkeypox and other endemic diseases as most Nigerian states laid emphasis on the surveillance and monitoring of the pandemic [30]. In addition, the fear associated with COVID-19 may have led to people not seeking care in health facilities, resulting in missed cases [30]. Current estimates show that there are currently 21 cases of MPVX and one death in Nigeria [5], which calls for concern, given that countries are lifting restrictions on COVID-19. Borders are reopened, it is no longer a requirement to wear masks outdoors in some places, and rules on social gatherings have been relaxed [31]. The surge in the number of exported cases of monkeypox may not be unrelated to these changes. However, it is unknown if COVID-19 restrictions impacted the spread of MPXV and warrant further research. Given the recent number of travel-related cases overseas and the current number of cases, it is evident that several factors may drive the outbreak. These include the waning herd immunity from smallpox vaccination which offered cross-protection for MPXV, contact with wild animals including the sale and consumption of bushmeat, poor community knowledge and lack of awareness of the disease [32,33].

## 3. Recommendations and Considerations for Future Research

Monkeypox is incorporated into Nigeria’s Integrated Disease Surveillance and Response System (IDSR), and a one health approach is currently being used to manage the outbreak [34]. However, there are gaps in knowledge about the epidemiology, host reservoir, transmission, role of environmental factors and genetic diversity of the disease [4,35,36]. Thus, Nigeria and other African countries need to increase the surveillance and detection of monkeypox cases to aid the understanding of this resurging disease [37]. While the global focus is on reducing the incidence of COVID-19, it may be worthwhile to consider a similar collaborative effort internationally to reduce the incidence of MPXV. It is vital to realize that diseases have no borders, especially with globalization and increased travel [4]. With the increasing number of travel-related outbreaks of monkeypox amid this pandemic, the burden of this crisis cannot be overemphasized. In addition, the outbreaks in other non-endemic countries emphasize the need for a concerted global effort focusing on the surveillance and rapid identification of cases, which are crucial for outbreak containment, while scaling up response to curb any further spread [38].

Furthermore, the dearth of knowledge and studies on monkeypox from Africa warrants the need for more research studies on monkeypox. Although there is no specific treatment for monkeypox infection, the present JYNNEOS vaccine licensed in the United States for monkeypox and smallpox should be deployed for targeted intervention in the high-risk regions [39].

## Figures and Tables

**Figure 1 viruses-14-01283-f001:**
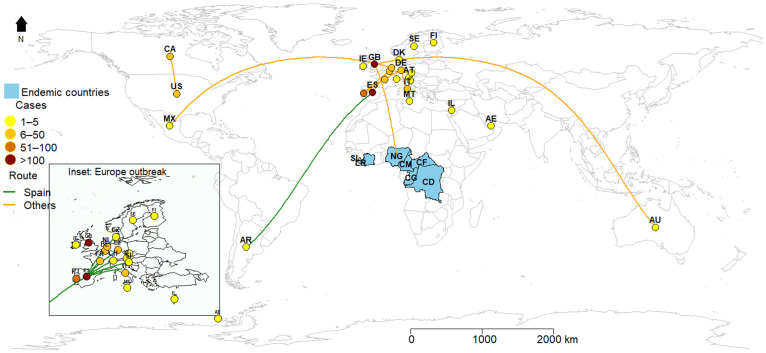
Geographical distribution of the ongoing monkeypox virus (MPXV) outbreaks. MPXV endemic countries have had sporadic epidemics since 2017. Nigeria (15), Cameroon (3), Central African Republic (8), Democratic Republic of Congo (1238), Sierra Leone (0). **Note:** Non-endemic countries: Argentine (AR), Australia (AU), Austria (AT), Belgium (BE), Canada (CA), Czech Republic (CZ), Denmark (DK), Finland (FI), France (FR), Germany (DE), Ireland (IE), Israel (IL), Italy (IT), Malta (MT), Mexico (MX), The Netherlands (NL), Portugal (PT), Slovenia (SI), Spain (ES), Sweden (SE), Switzerland (CH), United Arab Emirates (AE), UK (GB), USA (US). Endemic countries: Cameroon (CM), Central African Republic (CF), Democratic Republic of the Congo (CD), Liberia (LR), Nigeria (NG), the Republic of the Congo (CG) and Sierra Leone (SL).

**Table 1 viruses-14-01283-t001:** List of laboratory-confirmed cases of human monkeypox (MPX) outside of Africa in May, 2022. Few cases linked to West African clade ^†^.

Cases	Origin	Destination	Route of Transmission	Ref
1	Netherlands	Mexico	Resident New York, probably acquired in the Netherlands	[10,11]
1	Unknown	Malta	Patient recently travelled to a country with MPX cases	[10,12]
1	Europe	Finland		[10]
1	Unknown	Ireland		[10]
2	Spain	Argentina	Both cases had travelled to Spain	[10,13]
5	Belgium	Czech Republic	Index case had travelled to Antwerp, Belgium	[10]
2	Spain	Slovenia	Both cases had travelled to Spain	[10]
2	Spain	Denmark	Both cases had travelled to Spain	[10]
1	Unknown	Austria		[10]
4	Spain	Switzerland	Second confirmed case had travelled to Spain	[10]
2	Western Europe	Israel	Index case had travelled to Western Europe	[10]
26	Belgium	The Netherlands	Index case had travelled to Belgium	[10,14]
2	UK	Australia	Index case had a recent travel to the UK	[10,15]
22	Spain, Portugal	Germany ^†^	At least three cases linked to travel to Portugal, Spain or other regions of Europe	[7,10]
8	Lisbon, Portugal	Belgium ^†^	Index case had a recent travel to Portugal	[10,16]
16	Spain	France	Fourth confirmed case had travelled to Spain	[7,10]
12	Spain, Germany	Italy	At least four cases linked to travel to Spain or/and Germany	[7,10]
2	Unknown	Sweden	Unknown	[7,10]
14	Canada, Africa, Europe	USA	At least five cases linked to travel to West Africa, Canada, Europe	[7,10]
26	Unknown	Canada	Unknown	[7,10]
116	Unknown	Spain	Unknown	[7,10]
96	Unknown	Portugal ^†^	Unknown	[10,17]
106	Nigeria	UK	Index case had a recent travel to Nigeria	[6,10]

**Table 2 viruses-14-01283-t002:** Laboratory-confirmed cases of human monkeypox outside of Africa (2018–2021). NA = Not available.

Month/ Year	Cases	Possible Source of Infection	Origin	Destination	Clade	Route of Transmission	Secondary Transmission	Contacts Investigated	Ref
November,2021	1	NA	Nigeria	MD, USA	West African	Travel via Istanbul and Washington DC	No	NA	[21]
July, 2021	1	NA	Lagos, Nigeria	TX, USA	West African	USA-Nigeria-USA. Stayed in Lagos and in Ibadan, Oyo State	No	>200	[21]
May, 2021	3	1 case: unknown, 2–3 cases: human-to-human transmission	Delta, Nigeria	UK	West African	Two adults and one toddler. Travel via Istanbul, Turkey.	Yes (2 HH of index case)	38	[22]
December, 2019	1	NA	Nigeria	UK	NA	Very scarce information about this case.	Unknown	NA	[23]
May, 2019	1	Ingestion of barbecued bushmeat that might have been contaminated.	Nigeria	Singapore	West African		No	27 HCWs (close contacts)	[24,25]
October, 2018	1	Disposed of 2 rodent carcasses at his residence.	Nigeria	Israel	West African		No	5 HH + 11 HCWS	[26]
September, 2018	2	Case 1: contact with an individual with a monkeypox-like rash and consumption of bushmeat. Case 2: changing of potentially contaminated bedding.	Nigeria	UK	West African	Travel via Paris, France	Yes (HCW)	158 low risk, 125 intermediate-risk, 5 high risk	[27,28]
September, 2018	1	NA	Abuja, Nigeria	Cornwall, UK	West African	Travel from London to Cornwall by train	NA	NA	[27]

## Data Availability

Data used in this study are publicly available at https://github.com/globaldothealth/monkeypox (accessed on 15 December 2021).

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
