# Peer review of "Travel-Related Monkeypox Outbreaks in the Era of COVID-19 Pandemic: Are We Prepared?"

_viruses, 2022, doi:10.3390/v14061283_

Round 1

Reviewer 1 Report

The commentary addresses the recent reports on monkeypox virus outbreak detected and monitored in several countries. This is an ongoing investigation and sure will take additional time to know the extent and mode of infection. Only minor edits. The file is attached.

Author Response

Thank you for your comments. We have deleted the word "Human" from the Human monkeypox virus.

Reviewer 2 Report

This is a timely and important perspective piece. It is succinct and well written. I found the commentary to be well-researched with an appropriate blend of historical facts with present-day issues.

Some minor points for improvement:

line 37: do not capitalize "Outbreak"

line 39: do not italicize "events"

line 57: "70s" should be changed to "1970s"

line 64: strike the phrase "the dreaded"

line 86: "It's" should be changed to "It is"

line 92: there should not be a comma after the word monkeypox

lines 94-99: the author contributions sections is not completed properly.

Author Response

Thank you for your comments. Please find below our responses:

  1. line 37: do not capitalize "Outbreak" Response: We have revised it.
  2. line 39: do not italicize "events" Response: It has been revised.
  3. line 57: "70s" should be changed to "1970s" Response: It has been changed
  4. line 64: strike the phrase "the dreaded" Response: It has been deleted.
  5. line 92: there should not be a comma after the word monkeypox Response: Deleted.
  6. line 86: "It's" should be changed to "It is" Response: It has been changed
  7. lines 94-99: the author contributions sections is not completed properly. Response: We have added author's contributions

Round 2

Reviewer 2 Report

I have no additional comments. Best of luck!